

# Local processes with global impact: unraveling the dynamics of gas evasion in a step-and-pool configuration

Paolo Peruzzo[1], Matteo Cappozzo[1], Nicola Durighetto[1], and Gianluca Botter[1]

[1]Department of Civil, Environmental and Architectural Engineering, University of Padua, Italy.

**Correspondence:** Paolo Peruzzo (paolo.peruzzo@dicea.unipd.it)

**Abstract.** Headwater streams are important sources of greenhouse gases to the atmosphere. The magnitude of gas emissions originating from such streams, however, is modulated by the characteristic microtopography of the river bed, which might promote the spatial heterogeneity of turbulence and air entrainment. In particular, recent studies have revealed that step-and-pools, usually found in close sequences along mountain streams, are important hotspots of gas evasion. Yet, the mechanisms that drive gas transfer at the water-air interface in a step and pool configuration are not fully understood. Here, we numerically simulated the hydrodynamics of an artificial step-and-pool configuration to evaluate the contribution of turbulence and air entrainment to the total gas evasion induced by the falling jet. The simulation was validated using observed hydraulic features (stage, velocity) and was then utilized to determine the patterns of energy dissipation, turbulence-induced gas exchange, and bubble-mediated transport. The results show that gas evasion is led by bubble entrainment and is mostly concentrated in a small and irregular region of a few $\mathrm{dm}^2$ near the cascade, where the local gas transfer velocity, $k$, peaks at $500\ \mathrm{md}^{-1}$. The enhanced spatial heterogeneity of $k$ in the pool does not allow one to define *a priori* the region of the domain where the outgassing takes place, and makes the value of the spatial mean of $k$ inevitably scale-dependent. Accordingly, we propose that the average mass transfer velocity could not be a meaningful metric to describe the outgassing in spatially heterogeneous flow fields, such as encountered in step-and-pool rivers.

## 1 Introduction

Headwater streams play a pivotal role in the biogeochemical functioning of river networks and regulate key ecosystem services associated to fluvial environments (Appling et al., 2018; Bernhardt et al., 2018). Furthermore, mountain streams not only transport matter and chemicals from the uplands to downstream water bodies, but are also able to exchange gaseous compounds with the overlying atmosphere (Butman and Raymond, 2011; Battin et al., 2009). Despite the small flow rates and reduced water-air interfaces that characterize low-order streams, the efficiency of gas transfer in high-energy mountain settings makes these water bodies a significant source of greenhouse gases (such as $CO_2$ and $N_2O$ to the atmosphere. Thus, enhancing our understanding of gas evasion from headwater streams would have important implications for a broad range of fields of biology and environmental sciences (Kroeze et al., 2005; Crawford et al., 2014; Schelker et al., 2016; Marzadri et al., 2017; Horgby et al., 2019; Hall and Ulseth, 2020).





Yet, quantifying the cycling and release of greenhouse gases in low-order streams remains challenging, owing to the intertwined effect of different biotic and abiotic agents. In particular, the physical mechanisms responsible for gas evasion in river networks are often modulated by the hydrodynamic features of the flow field, which are in turn strongly impacted by the (micro)topography of the river bed (Duvert et al., 2018; Botter et al., 2021; Rocher-Ros et al., 2019; Looman et al., 2021). Differently from floodplain rivers - that exhibit smooth hydraulic conditions - the steep slopes and irregular beds typical of

mountain streams yield to complex and heterogeneous flow patterns. In this setting, the free surface of the water flow can frequently break, owing to obstacles, bends or abrupt variations of the channel bottom.

Recent studies evidenced that local steps are important hotspots of gas evasion, to the point that in high energy streams, where steps are distinctive of the stream's morphology, they control the overall network-scale outgassing (Vautier et al., 2020; Botter et al., 2022). While it has been argued that gas transfer in correspondence of riverbed drops is promoted by the concurrence of

high turbulence and bubbles entrainment (see e.g. Ulseth et al., 2019), there is limited knowledge of the physical processes that drive outgassing dynamics in local steps (ee e.g. Cirpka et al., 1993). Moreover, to date, the spatial patterns of energy dissipation and gas exchange in the plunging jet and the receiving pool of individual steps have not been analyzed. Consequently, the characteristic length-scales of turbulence-induced and bubble-mediated gas transport in a step and pool configuration are still unknown. The characterization of the local and spatially-heterogeneous nature of gas evasion in step and pool formations has

important practical and theoretical implications. In particular, the spatial heterogeneity of gas exchange along a step constraints our ability to conceptualize such geomorphic elements by identifying *a priori* the size of the surface through which most of the gas exchange occurs (i.e., the step gas footprint) and the corresponding "effective gas transfer velocity" therein.

The present work aims at filling these gaps of knowledge by combining experimental data and numerical modeling. In particular, here we reproduce the hydrodynamics of a simple step-and-pool geometry within an artificial flume using numerical

simulations, which are then used to model the underlying gas exchange mechanisms taking place across the step. The numerical analysis allows us to $i)$ quantify the spatial variability of the gas transfer velocity, discerning the contributions of the turbulence and the entrained in-water bubbles, and $ii)$ identify the spatial patterns of energy dissipation and gas evasion in a step-and-pool. The major implications of our findings for future gas transfer studies are also discussed.

## 2    Methods

### 50    2.1    In flume experiment

An experiment to study the hydrodynamics of a step-and-pool was carried out under controlled hydrodynamic conditions inside a horizontal, 6 m long plexiglass-made artificial flume with a rectangular section of width and height equal to 0.3 m and 0.5 m, respectively. The water circulates through the channel via a constant head tank that maintains a steady discharge of $2\,\mathrm{ls^{-1}}$, which is accurately measured by a magnetic flowmeter.

A step-and-pool formation was artificially created into the flume by inserting a broad-crested weir and a tapered positive step (see Figure 1). The height of the two elements was 25 and 9 cm, respectively. Owing to the difference in the water level upstream and downstream of the weir, a local step with a water height drop $\Delta H$ equal to $\approx$15 cm was obtained. The pool,





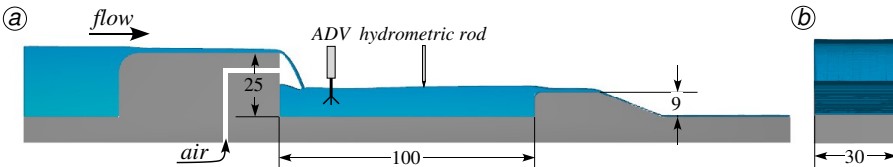

**Figure 1.** Schematic of a the step-and-pool. a) lateral and b) frontal view of the flume. Lengths are expressed in cm.

bounded by the weir and the step, had depth $z_0 \approx 11$ cm. The pool was long enough (100 cm, about ten times the water depth) so that the hydrodynamics near the jet was not impacted by the presence of the downstream positive step. Finally, an orifice on

the wall of the weir ventilated the air below the cascade jet to maintain atmospheric pressure therein.

We measured the water level along the step-and-pool using a graduated hydrometric rod (accuracy $\pm$ 0.1 mm) mounted on a moving carriage. Measurements were taken every 2 cm along the middle axis of the flume, allowing us to reconstruct the free surface profile in the step-and-pool.

Local flow velocity was assessed in some representative sections along the middle axis of the pool by two Acoustic Doppler

Velocimeters (ADVs). To this aim, we used a Vectrino Profiler ADV (Nortek, Rud, Norway) to record the velocity near the bottom. The profiler provides observations of the three-component velocity, $u_i$, with a resolution as fine as 1 mm over a 30 mm range placed 40 mm below the transmitter, and an output rate of 100 Hz. For each section, we carried out a series of measures as follows: starting from the bottom, we repeatedly raised the ADV by 10 mm increments, recording velocities for 30 mm of water column each time. Since the ADV works correctly only when the transmitter is dipped in the water, the fluid volume

close to the air-water interface (with a distance from the bed of more than 70 cm) was not analyzed. Therein, the velocity was assessed by a 2D SonTek ADV (Xylem Inc., Rye Brook, NY, USA). This ADV provides two-component velocity measures for a single point that corresponds to a control volume of 1 cm$^3$, with a maximum sampling rate of 50 Hz. Measures of the horizontal components of the velocity were taken every 10 mm for 70 mm starting from the free surface.

To provide statistically consistent samples, the ADVs recorded the velocity signal for more than 30 s. During the tests, talcum

powder was added to the water to achieve a Signal-to-Noise Ratio (SNR) higher than 30 dB, thus minimizing the background noise recorded by the instruments.

## 2.2    Numerical simulation of the step and pool

The in-flume experiment was numerically simulated using the software FLOW-3D Hydro 2022R1 (Flow Science Inc., Santa Fe, NM, USA), which adopts grids based on structured orthogonal cells to shape complex geometries using the fractional

area/volume method (FAVOR). The evolution of the free surface is instead modeled using the volume-of-fluid (VOF) method (Hirt and Nichols, 1981), which can also track complex interfaces.

To simulate the hydrodynamic field of the step and pool, the software solves the Reynolds-Averaged Navier-Stokes equations along the Cartesian axes $x_i$ coupled with the RNG $e_k - \varepsilon$ scheme ($e_k$ and $\varepsilon$ being respectively the turbulent kinetic energy per unit of mass and the turbulent kinetic energy dissipation rate) that resolves the turbulence at the sub-grid scale.





The simulated dynamics in the pool included a full description of the bubble entrainment associated to the water flow. The air model used in this paper describes air transport in the water column according to the standard Advection-Diffusion Equation of a scalar with known concentration $C$. In the simulation, the air bubbles were assumed to be are spheres with a radius, $r$, equal to 0.5 mm, a value congruent with the analysis of cam-derived images taken during the experiment and in line with previous experimental studies (Woolf et al., 2007; Klaus et al., 2022; Karn et al., 2016). The model allowed us to quantify the entrained

and escaping bubbles fluxes across the free surface, $q_B$.

In the numerical experiment, the cell size ranged from 2 mm in the region where the jet plunged to 8 mm upstream of the broad crest weir. In total, the flume was discretized using approximately 7.5 millions of cells. The boundary conditions were the following: i) a steady inflow of 2.0 ls$^{-1}$ upstream of the weir and ii) free outflow downstream of the step. The no-slip velocity constraint was set on both the bottom and channel sides, where the equivalent roughness was set as $\delta_e$=5×10$^{-2}$ mm.

Initially, fluid was at rest, and transient conditions were observed for less than 20 seconds before stationary conditions were reached in the domain.

### 2.3    Modeling gas exchange in step-and-pools

The gas exchange between water and the atmosphere takes place through the thin separation layer at the interface between the two fluids. The exchanged volumetric gas flux, $F$, is proportional to the difference in gas concentration between air and water

and is described by the Henry's law as

$$F = k \left( C_w - \frac{C_a}{K_H} \right) , \qquad (1)$$

where $k$ is the gas transfer velocity (representing the water depth equilibrated with the atmosphere in the unit of time), $C_w$ and $C_a$ are the gas concentration in water and atmosphere, respectively, and $K_H$ the Henry constant determined for the specific exchange process under investigation. When both fluids are at rest, the exchange rate depends on the molecular diffusivity of the

dissolved gas, $D$, which slowly drives the absorption to (or the release from) the water column. Instead, in turbulent flowing water $k$ is much higher than that induced by the molecular diffusivity. In this case, the eddies originating from turbulence continuously renew the water close to the free surface, enhancing the transfer of gas at the interface (Lamont and Scott, 1970; Katul and Liu, 2017). Hereafter, the gas transfer velocity driven by turbulent mechanisms of this type is defined as $k_T$.

However, in many real-world settings, air entrainment and bubbles can intensify gas exchange in heterogeneous flow fields,

primarily by increasing the total exchange area between water and air. The bubble-mediated gas exchange is here encapsulated by the bubble-mediated gas transfer velocity, $k_B$. Therefore, in turbulent flowing waters where air entrainment takes place, the total gas transfer velocity $k$ can be computed as the sum of velocities due to free-surface and bubble-mediated exchange (Klaus et al., 2022):

$$k = k_T + k_B . \qquad (2)$$



The separation of the contributions of $k_T$ and $k_B$ to the total gas exchange in a step-and-pool is one of the main contributions of this paper. The following subsections describe how these terms are computed in our numerical simulation.

### 2.3.1   Turbulence-induced air-water gas exchange in the flume

In mountain streams, with low water depths and high velocities, the theoretical schemes that are better suited to quantify $k_T$ are those based on small-eddies models (Moog and Jirka, 1999b), according to which $k_T$ is given by

$$k_T = \alpha_T \, Sc^{-0.5} \left(\nu \varepsilon\right)^{0.25} , \qquad (3)$$

where $\alpha_T$=0.2÷0.4 is a calibration factor, $\nu \approx 1\times 10^{-6}$ m$^2$s$^{-1}$ is the kinematic viscosity of water, and $\varepsilon$ represents the turbulent kinetic energy dissipation rate per mass unit close to the free surface. Moreover, in Equation (3), $Sc = \nu/D$ is the Schmidt number, which expresses the link between kinematic and molecular diffusivity. The nature of the processes described by Equations (1) and (3) is inherently local and occurs at the Batchelor scale, which is defined as $\lambda_B = \eta/\sqrt{Sc}$, where $\eta = (\nu^3/\varepsilon)^{1/4}$ is the Kolmogorov length, i.e., the characteristic length of the micro vortexes. In running waters, being $Sc > 1$, the spatial scale of the gas transport is thus smaller than that of the smallest eddies generated by turbulence in its energy cascade.

    Previous experimental studies have demonstrated that the exponent $-0.5$ in Equation (3) applies to cases in which the free surface is smooth, while the same exponent may decrease to $-0.67$ whether the free surface is riffled (Zappa et al., 2007). It should be noted that, in the general case, the gas exchange also depends on the fluid temperature, which affects both $\nu$ and $D$. However, since the present work aims at reproducing a short portion of a stream in which the water temperature is nearly constant, we neglect the thermal effect on $D$ and $\nu$, so that $Sc$ is assumed to be constant and the gas transfer velocity $k_T$ scales with $\varepsilon^{0.25}$.

    Nevertheless, in-field estimations of gas transfer velocity are usually performed at the reach-scale based on spatially-averaged quantities under the assumption of uniform flow. Under the above circumstances, the mean turbulent kinetic energy dissipation rate $\overline{\varepsilon}$ (hereinafter the overline applies to spatially-averaged quantities) can be expressed as:

$$\overline{\varepsilon} = g\overline{U}\,\overline{S} , \qquad (4)$$

with $g$ the gravity acceleration, $\overline{U}$ the cross-sectional flow velocity, and $\overline{S}$ the average bed slope. It should be noted the proposed scheme is suitable whether the turbulence generated at the bottom is the main driver of the overall intensity of the turbulence in the water flow. In headwater streams, instead, riverbed heterogeneity, partially emergent boulders, and small cascades can be important additional turbulence sources (Moog and Jirka, 1999a; Botter et al., 2022). In the particular setup analyzed in this study, the plunging jet of a cascade dissipates its energy by inducing high $\varepsilon$ in the outermost fluid layer, thereby preventing the use of equation (4).

    Consequently, $k_T$ is calculated based on the computed value of $\varepsilon$ near the free surface, using Equation (3) with $\alpha_T$=0.4 and standardizing the result with a Schmidt number equal to 600.





### 2.3.2 Bubble-mediated gas exchange

In this work, while dealing with bubble-mediated gas exchange we focus on the so-called "kinematic bubble effect" (Liang et al., 2013).

The magnitude of the kinetic bubble effect is strictly related to the bubble residence time (or bubble lifetime), $T_B$. A reasonable estimation of this residence time is given by

$$T_B = \alpha_B \frac{z_0}{u_B},\tag{5}$$

where $\alpha_B$ is an $O(1)$ coefficient and $u_B$ is the bubble rise velocity. Here, $u_B$ was estimated using the model proposed by Woolf (1993):

$$u_B = \begin{cases} 0.172\, r^{1.28}\, g^{0.76}\, \nu^{-0.56} & \text{if } r \leq 0.82\, mm \\ 0.25\, ms^{-1} & \text{if } r > 0.82\, mm \end{cases}\tag{6}$$

During the time $T_B$, any bubble is able to exchange gas with the surrounding water as a function of $u_B$ and the Reynolds number $Re_B = 2\, u_B\, r / \nu$. In particular, the velocity of gas exchange, $j$, through a bubble is given by the following expression (valid for $Re_B > 10$):

$$j = \sqrt{\left(1 - \frac{2.89}{\sqrt{Re_B}}\right) \frac{2 D\, u_B}{\pi\, r}}.\tag{7}$$

The velocity of gas exchange is functional to the computation of the gas equilibration time constant of the bubble, $T_g$, which defines the time-scale of the gas transfer process and is given by:

$$T_g = \frac{r}{3\, j\, \beta},\tag{8}$$

where $\beta$ is the Ostwald solubility coefficient.

The gas transfer velocity induced by bubbles, $k_B$, can be linked to the ratio between $T_B$ and $T_g$ assuming a first-order reaction model which accounts for the time-dependent changes of gas concentration within the drifting bubble:

$$k_B = \frac{q_B}{\beta} \left(1 - e^{-\frac{T_B}{T_g}}\right),\tag{9}$$

with $q_B$ the bubbles flux per unit area.

In this application, $k_B$ was calculated by combining the above equations, as detailed in what follows. The bubble rise velocity was calculated assuming a radius $r$ of 0.5 mm using Equation (6) (leading to $u_B = 0.133\ \text{ms}^{-1}$). Then, the bubble lifetime $T_B$





was calculated setting $\alpha_B = 1.0$ in Equation (5) ($T_B \approx 0.83$ s) and the Reynolds number was computed based on its definition ($Re_B = 133$). After that, assuming that in the case of $CO_2$ $D = 1.6 \times 10^{-9}$ $m^2 s^{-1}$ and $\beta = 0.94$, the bubble's transfer velocity, $j$, and the equilibration time constant, $T_g$, were estimated via Equations (7) and (8) ($j \approx 0.5$ $mms^{-1}$ and $T_g = 0.34$ s). Finally, $k_B$ was obtained from Equation (9) using the local flux $q_B$ computed by the numerical code. Given the uncertainty in the value of $T_B$ in this particular experiment, a range of values for the gas transfer velocity induced by bubbles was derived by varying $\alpha_B$ in the interval (0.5,2.0).

## 3 Results

### 3.1 Performance of the hydrodynamic model

To assess the reliability of the hydrodynamic model, the depths and velocities measured in the pool were compared to the numerical solution provided by the code. Figure 2a shows the comparison between the observed and modeled water levels along the middle axis of the flume. The water stage ranged from 10.5 in the pool to 12.0 cm downstream of the water jet, whereas the backwater upstream of the plunge had higher water depths. The modeled free surface fitted the experimental levels in the pool, over the broad-crested weir, and, with lower accuracy, on the cascade. Therein, however, the measures taken with the rod were sub-optimal owing to observed abrupt variations in the water level. Further, Figure 2a shows the model ability to reproduce the module of the velocity vector, $|\boldsymbol{U}|$. The results highlight a non-uniform flow distribution along the vertical direction within the pool and downstream of the cascade.

Panel b of Figure 2 reports the profile of the time-averaged longitudinal velocity $\langle u_x \rangle$ in three sections located at 10 (A), 20 (B), and 30 cm (C) downstream of the plunging jet. The velocities estimated by the ADVs campaign were rather scattered, likely because of the strong fluctuations induced by the turbulent vortexes, and systematically underestimated close to the bottom. Nevertheless, the observed trends were consistent with the numerical solution of the hydrodynamic model. In all cases, the maximum values of $\langle u_x \rangle$ were close to the bottom and swiftly decreased downstream of the cascade (from approximately 0.75 $ms^{-1}$] (A) to 0.45 $ms^{-1}$ (C)). Moreover, the $\langle u_x \rangle$ decreased along the vertical direction, with a null velocity at 0.45 $z_0$ for Section A and 0.5÷0.55 $z_0$ for Section B and C. Negative values are observer at a higher vertical distance from the channel bed, $z$. Consequently, the maximum backflow was observed near the free surface.

The three examples of velocity profiles pointed out the persistence of a large, counterclockwise, eddy structure. The jet plunging in the pool also formed a clockwise vortex in the backwater region, as clearly indicated by the streamlines reported in Figure 2c. During the experiment, air bubbles entrained by the water jet were visible within a 30-cm wide portion of the counterclockwise eddy and in the backwater region (blue circles in panel d of Figure 2). The size of most of the bubbles highlighted in the picture lay in the range $r = 0.5 \div 1.0$ mm, while smaller bubbles were hardly feasible to detect.





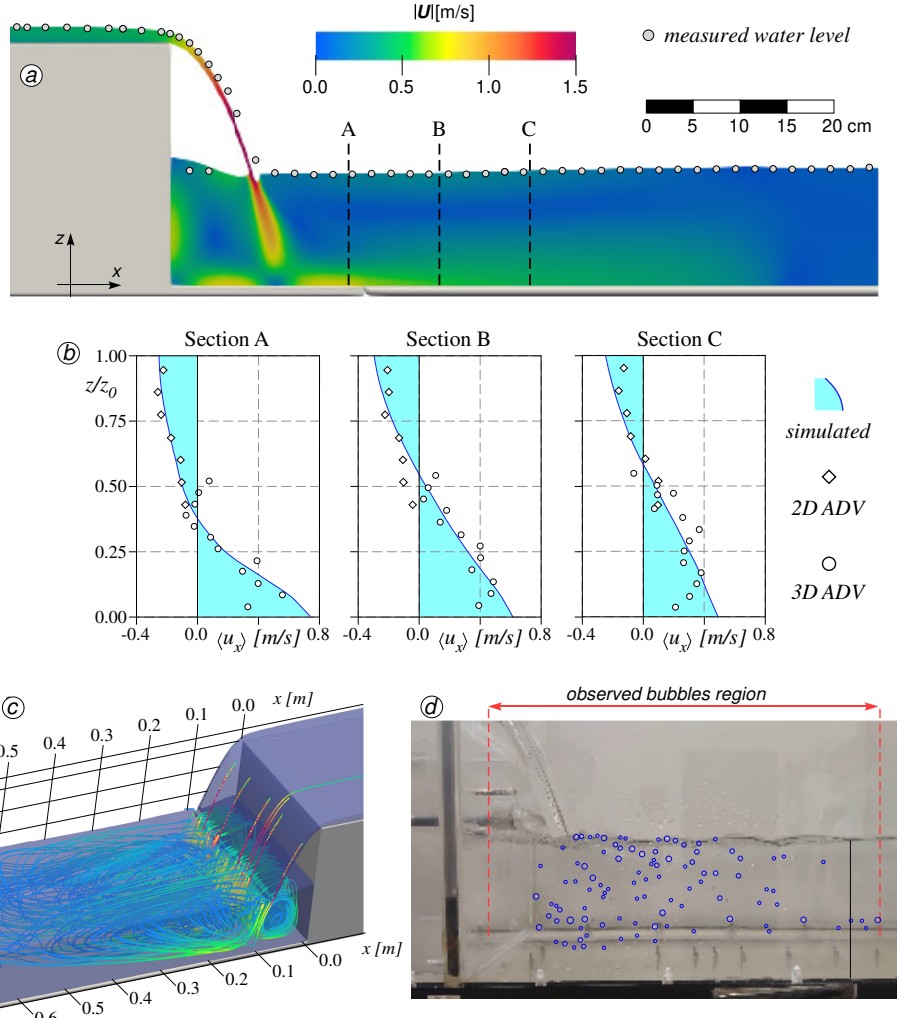

**Figure 2.** Step-and-pool hydrodynamics. a) Free surface computed by the FLOW-3D simulation vs that measured by the hydrometric rod (grey circles) and field of the module of the time-averaged velocity, $|U|$ in the middle axis of the pool. b) Time-averaged longitudinal velocity profile, $\langle u_x \rangle (z)$, in three sections of the pool; solid blue line with cyan area denotes the numerical solution, circles and diamonds 3D and 2D ADV measures, respectively (for the sake of clarity, ADV measures are reported every 10 mm). c) Streamlines in the first portion of the pool colored according to $|U|$, d) Bubbles observed in the pool (blue circles).

## 3.2 Gas evasion produced by the step-and-pool configuration

The simulated spatial patterns of total turbulent kinetic energy dissipation rate were highly heterogeneous (Figure 3a). As expected, the water spilled from the weir dissipated almost all its energy within the region of the two large eddies. However, 200 the highest values of $\varepsilon$ were found in the area where the water jet plunges. Therein, $\varepsilon$ was higher than 2.0 $\mathrm{m^2 s^{-3}}$ at the free





surface, with much smaller values - of the order of 0.1 m$^2$s$^{-3}$ - near the bottom of the flume. The decrease of $\varepsilon$ with the depth was more than linear (see inset of Figure 3a). Therefore, overall, the first centimeters of the water column dissipated a significant amount of the jet energy. Moreover, the simulated patterns of $\varepsilon$ indicated that, in a small superficial region of the fluid mass with a size of 120 cm$^2$ (that is comparable with the footprint of the cascade on the pool), the energy dissipation rate was one order (at least two orders) of magnitude larger than that estimated in the backwater region (downstream of the cascade).

The recirculating zones were also able to exchange gas with the overlying atmosphere as they trapped the air dragged by the falling jet (see Figure 3b). The two large eddies confined the air in some low-velocity regions located in the middle of the water column and swept the bubbles near the bottom and the free surface. According to the numerical model, the maximum bubble concentration was found close to the plunging jet ($C \approx 50$ mgl$^{-1}$) and only a small amount of bubbles was predicted at the edges of the region of the observed bubbles. Despite the steady flow condition achieved (the global air mass in the pool remained almost constant during the simulation), the air entrainment spatial distribution varied significantly over time (see the video in Supplemental Material). The magnitude of bubbles flux, $q_B$, mirrored the patterns of air concentration (Figure 3c). $q_B$ was highly heterogeneous in the pool, and most air bubbles escaped by bursts in big spots within the first 10 cm downstream of the cascade and the first 5 cm of the backwater region. In contrast with $\varepsilon$, the maximum value ($q_B \approx 5.0$ mms$^{-1}$) was not found in the correspondence of the cascade, where the air was entrained, but a few centimeters downstream of it.

The gas evasion induced by the step resulted from the superposition of the effect of turbulence and entrained bubbles. Figure 4a shows the spatial distribution of $k_T$. We also averaged $k_T$ along $y$-direction ($\overline{k}_T$) and reported its longitudinal pattern along the step-and-pool. Owing to the close connection between the gas exchange velocity and $\varepsilon$, $\overline{k}_T$ achieves the maximum of 50 md$^{-1}$ near the plunging jet. In this region of enhanced dissipation, local $k_T$ values peaked at 100 md$^{-1}$; such a value was almost one order of magnitude larger than those observed downstream of the pool, where $k_T$ progressively decreased moving away from the jet. Conversely, in the backwater region $k_T$ was nearly uniform and $\overline{k}_T$ did not exceed 20 md$^{-1}$.

The bubble-mediated gas transfer, $k_B$, is shown in Figure 4b. As for $k_T$, we also estimated the longitudinal distribution of the transverse-averaged exchange velocity $\overline{k}_B$.

In the region of the domain in which the bubble flow was enhanced, we estimated for $k_B$ values which were almost an order of magnitude higher than those computed for the turbulence-driven gas transfer velocity, $k_T$. In particular, $k_B$ locally peaked to 500 md$^{-1}$, while the highest values of $\overline{k}_B$ exceeded 100 md$^{-1}$ - about twice the maximum value of $\overline{k}_T$.

The patterns of $k_T$ and $k_B$ were not strongly correlated in space and both showed pronounced peaks. Consequently, the relative contribution of bubbles and turbulence to the total gas evasion showed a heterogeneous spatial trend. Figure 4c reports the pattern of the dominance ratio $\chi = \overline{k}_B/\overline{k}_T$ along the pool where bubbles entrainment was observed. The plot indicates that bubbles dominated the exchange rate ($\chi > 1$) in the central portion of the bubble region. In contrast, turbulence led the outgassing ($\chi < 1$) near the cascade (where $\varepsilon$ was maximum) and at the edges of the pool (where air concentration was low).



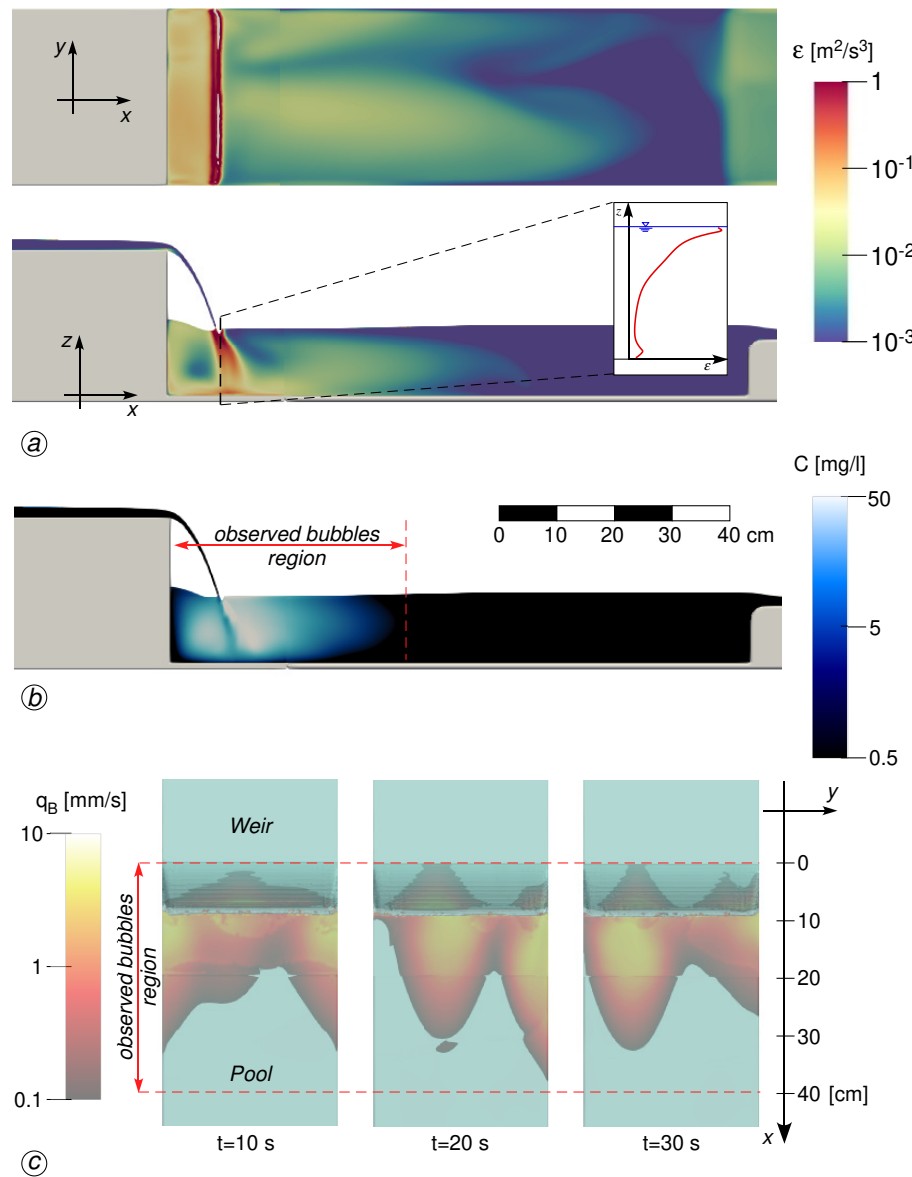

**Figure 3.** Scalar fields in the pool. a) Planar (top) and lateral (bottom) view of the turbulent kinetic energy dissipation rate, $\varepsilon$. b) Lateral view of the air concentration. c) Top view of outgas flux, $q_B$ at three simulation times. Lateral views are in the middle axis of the pool. The right inset in a) shows the computed $\varepsilon$ along $z$ in correspondence with the plunging water jet.

## 4    Discussion

The important contribution of headwater streams to the global emissions of climatically relevant gases is modulated by the

topography of the river bed, which determines the steep, tortuous, and irregular nature of water flow paths in high-energy



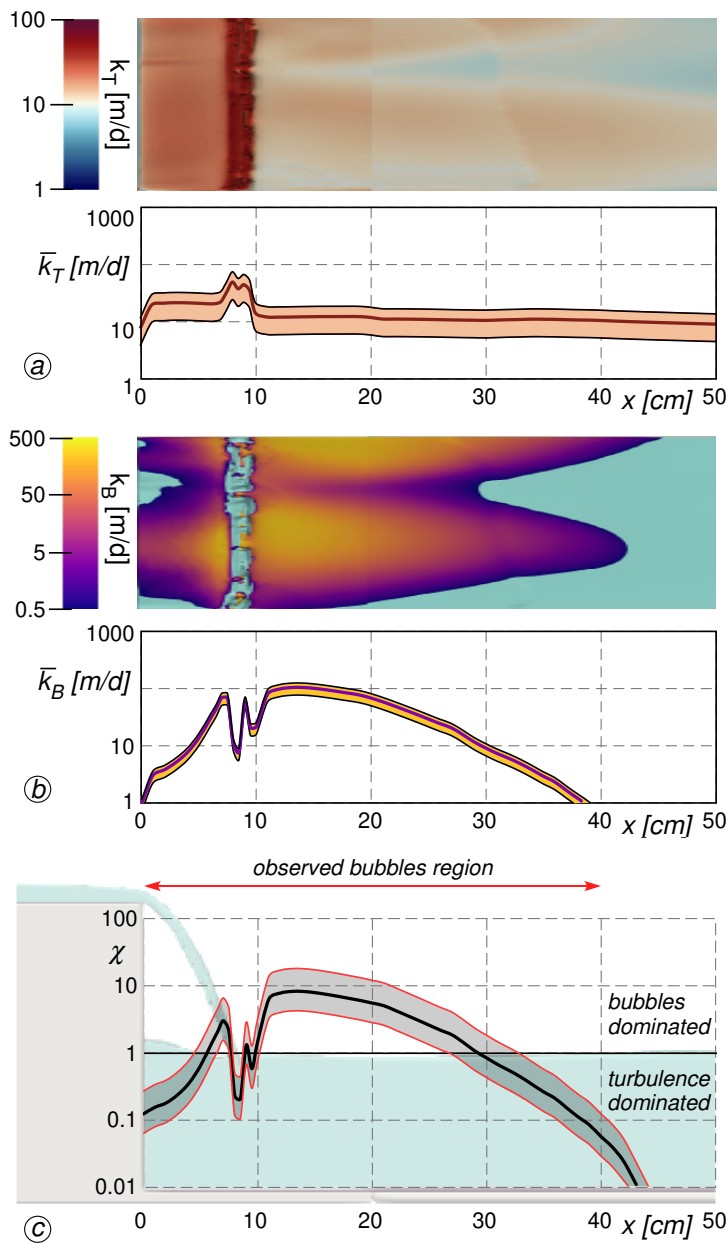

**Figure 4.** Gas exchange velocity in the pool. a) Field of $k_T$ and transverse-averaged exchange velocity $\overline{k}_T$. The solid line is estimated by Equation (3) with $\alpha_T = 0.4$, while the range of values (colored area) is determined by assuming $\alpha_T$ equal to 0.2 and 0.6. b) Field of $k_B$ and transverse-averaged exchange velocity $\overline{k}_B$. The solid line is estimated by Equation (9) with $\alpha_B = 1$, while the range of values (colored area) is determined by assuming $\alpha_B$ equal to 0.5 and 2. c) Ratio $\chi = \overline{k}_B / \overline{k}_T$ within the pool.





settings (Duvert et al., 2018; Wallin et al., 2011; Marx et al., 2017). For example, small-scale heterogeneity of channel forms promotes rippled air-water interfaces that enhance the evasion of greenhouse gases. The sequences of step-and-pools typical of mountain streams are plastic examples of small-scale geomorphic elements of rivers that control gas exchange processes at the water-air interface, eventually bearing a relevant impact on the chemical equilibrium of the atmosphere (Botter et al.,

2022). Sharp discontinuities of the river bed of the types observed in steps and small cascades cause a sudden transfer of potential energy to kinetic energy of the water flow. The water jet impacting the pool or the downstream portion of the riverbed then loses a large amount of its energy, roughly corresponding to the height difference between the water levels upstream and downstream of the cascade (i.e., $\Delta E \approx g\,\Delta H$, where $\Delta E$ is the dissipated energy per unit of fluid mass, see Botter et al. (2022)). The dissipation of energy occurs within a few centimeters - a very short spatial scale as compared to the characteristic

length of a uniform stream reach with the same energy losses (1÷10 m).

In the pool located downstream of a step, the loss $\Delta E$ occurs locally through the production and the subsequent dissipation of turbulent kinetic energy. Our experiment highlights that the $\varepsilon$ is strongly nonlinear and rapidly decreases downstream of the plunging jet and towards the bottom of the pool. The overall result is that the turbulence depletes indicatively the 15÷20% of $\Delta E$ near the surface within a length that does not exceed a few centimeters nearby the cascade.

In addition, the velocity increase observed within the falling jet is responsible for the entrapment and entrainment of the air that overlies the pool. The air bubbles in the flume experiment extended for almost 30 cm downstream of the jet. However, the numerical simulation indicates that most of the air is exchanged only in the first 10 cm near the jet.

In this area, the overall transfer velocity $k = k_{T} + k_{B}$ achieves a local maximum of about 500 md$^{-1}$, with transverse-averaged values of the mass transfer rate equal to $\bar{k} = 120$ md$^{-1}$. While we recognize that such values of $k$ are quite high

- especially in the light of the uniformity of the geometry in our flume - we expect that the outgassing induced by steps in mountain streams could be even larger owing to the higher values of $\Delta H$ (Natchimuthu et al., 2017; Schneider et al., 2020; Botter et al., 2022) and the enhanced heterogeneity of bed geometry, which arguably intensify turbulence and air entrainment (Vallé and Pasternack, 2006; Vautier et al., 2020).

Interestingly, in the bubbles region of the pool, the potential outgassing, i.e., the surface integral of Equation (1) for a constant

and unit air-water concentration difference, induced by bubble-mediated transport is 2.25 times higher than the corresponding outgassing implied by turbulence alone. Although this analysis is based on the assumption of a constant and uniform bubble radius, we believe that the orders of magnitude of the underlying processes are properly captured by the model, as space-time heterogeneity in bubble size should not impact significantly the estimated flux of entrained air. Further, we suggest that this result could apply also to different settings, as white waters are observed not only in correspondence of local steps but also in

turbulent reaches where high Froude numbers, macro roughness, and emerging stones can promote diffusive entrainment of air. Accordingly, we propose that most of the observed gas evasion in high-energy streams is likely bubble-mediated. Therefore, we suggest that more efforts should be directed to adequately model entrained air and gas transfer through bubbles in high energy streams, where channel bottom morphology causes rippled-free surfaces.

Moreover, the key role of the bubbles in gas evasion from high-energy streams suggested by our results raises two issues

about the correct procedure to define the value of $k$ for a given stream reach. First of all, our numerical simulation indicates



that the standardization of $k$ should be done not only in terms of $Sc$ but also in terms of $\beta$ (and $D$), as suggested in previous studies (Woolf et al., 2007; Hall and Ulseth, 2020; Klaus et al., 2022). Consequently, some caution is necessary when one gas evasion measurements are used under the assumption $k \approx k_T$ if tracer gases with different solubility are employed. For instance, whether in the present analysis low solubility gases (such as $O_2$ and He) were considered, the ensuing $k_B$ values would be, respectively, 3.0 and 4.3 times higher than that estimated for $CO_2$ (as $\beta_{O_2} = 0.034$ and $\beta_{He} = 0.094$). Therefore, the effect on the bubble mediated gas exchange induced by changes in solubility is much higher than that captured by a simple $Sc$ scaling.

Second, in spite of the observed empirical correlation between $k$ and $\overline{\varepsilon}$ in high-energy streams (Ulseth et al., 2019), our results indicate that the definition of a robust upscaling law between the gas transfer rate and the turbulent kinetic energy dissipation rate needs to include the spatial frequency of steps and the relative contribution of bubble mediated and turbulent induced gas evasion, as long as the scaling of $k_B$ could be modulated by local hydrodynamics features such as the flow velocity, the water depth, and the channel morphology.

A final remark concerns the local nature of the outgassing process in our step-and-pool and the related implications for future gas transfer studies. Here, 75% of the bubble-mediated potential outgassing occurs through spots with high $q_B$, the area of which does not exceed 2.5 dm$^2$ overall - less than 20% of the region where air entrapment is visible. This heterogeneity of gas evasion in the pool is also mirrored by the spatial patterns of $k_T$. The first 5 cm downstream of the cascade caused almost 25% of the potential outgassing induced by turbulence in the whole domain. Thus, both the turbulence and bubble-mediated processes mostly take place close to the plunging jet and operate at a scale that is much smaller than the length used to measure $k$ during in-field experiments, which ordinarily adopt probes or chambers with an inter-spacing of at least some meters (Vingiani et al., 2021; Vautier et al., 2020; Botter et al., 2022). This difference of scale might explain the heterogeneity of $k$ that can be empirically-estimated from tracer gas concentration drops observed along reaches or segments that contain steps or cascades (Wallin et al., 2011; Natchimuthu et al., 2017; Leibowitz et al., 2017). In these instances, $k$ is implicitly averaged over the entire fluid region included within the reference sampling points and cannot capture the local peaks of $k$, as the step contribution to the outgassing is eventually "smoothed" on a larger area. An important consequence of the local and heterogeneous nature of the outgassing process in steps and pools is that, in cases where the spatial patterns of local $k$ are unknown, the measure of the transfer velocity is dependent on the position of the monitoring points and the geometry of the steps. Thus, the average value of $k$ across a step is not only unpredictable but also inevitably scale-dependent, i.e., the larger the water volume involved in the averaging, the lower the effective mass transfer velocity $k$ (Botter et al., 2021, 2022). Based on the above arguments, we propose that the use of the mass transfer rate, $k$, should be dismissed in cases in which the heterogeneity of the flow field controls the fraction of mass evaded into the atmosphere, as in our step-and-pool configuration.

## 5 Conclusions

In this study, we have numerically simulated the hydrodynamics of a step-and-pool, which was also reproduced in a laboratory-designed flume experiment. Computational values of key hydrodynamic parameters that control the gas transfer between water
and the atmosphere were then used to estimate the spatial patterns of gas transfer velocity, and the contributions to the gas

evasion due to turbulence and air entrainment. Our simulation shows that the cascade downstream of the step dissipates energy and entraps air in a small, irregular region near the chute (about 3 $\mathrm{dm}^2$), in which the mass transfer velocity $k$ is locally very high (500 $\mathrm{md}^{-1}$). This result indicates that gas evasion in step and pool configuration may be a very local process, taking place in a few $\mathrm{dm}^2$ close to the plunging jet. The numerical simulation also suggests that bubble-mediated gas exchange dominates the turbulence-induced outgassing, reinforcing the idea that gas evasion in mountain streams could be mainly driven by bubbles

and white waters. The observed heterogeneity of $k$ - the pattern of which is linked to the specific geomorphic features of the step under investigation - raises concerns about the ability of traditional metrics such as the mass transfer rate to quantify gas emissions from step-and-pools and morphologically-heterogeneous streams.

*Data availability.*   Data supporting the findings of this study are openly available in Peruzzo et al. (2023) at http://researchdata.cab.unipd.it/id/eprint/619.

*Video supplement.*   Animation of air entrainment is included with the paper.

*Author contributions.*   Conceptualization: PP, ND, GB. Methodology: PP, MC, GB. Investigation: MC. Data Curation: MC. Software: PP. Formal Analysis: PP, GB. Visualization: PP. Writing - original draft preparation: PP. Writing - review and editing: ND, GB. Funding: GB. Supervision: GB.

*Competing interests.*   The authors declare no competing interests.

*Acknowledgements.*   This research was supported by the European Community's Horizon 2020 Excellent Science Programme (grant no.
H2020-EU.1.1.-770999).





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
