# Peer review of "Local processes with global impact: unraveling the dynamics of gas evasion in a step-and-pool configuration"

_Biogeosciences, 2023_

## Author Comment (AC2)

Dear Reviewer,

Thank you so much for your valuable comments, and the positive assessment of our paper. Please find below a preliminary answer to your comments.

Best regards,

Paolo Peruzzo (on behalf of all authors)

The article by Peruzzo et al explores the gas exchange dynamics in a step-and-pool situation, which is a common geomorphic feature in many streams with some relieve like mountain streams. To do this they use a lab setup, with a unique monitoring set that measures water velocity and turbulence in multiple directions and at a high spatial and temporal resolution. This is something I have rarely seen so the authors can explore experimentally at a great resolution the drivers of gas exchange in waters. They can also separate bubble mediated gas exchange which is important ad often ignored.

Thank you for the nice summary of our work.

They find a high spatial in the gas exchange, both turbulence and bubble mediated, with a very small area accounting for most of the gas exchange with the atmosphere. The article is well written and figures of high quality…

Thank you for the positive assessment of our manuscript.

…but the technical level of some of the section may be hard to follow for not only the average reader of "Biogeosciences", but also for someone who works with gas exchange in rivers without a hydrologic engineering background like me. With this I provide some suggestion to clarify some terms, notations, and provide citations on those technical aspects of the work.

Thank you for your comment. We will work to improve the readability for the average readership of Biogeosciences starting from your suggestions. In particular we will revise Section 2.3 including additional references and explaining the most technical aspects related to the hydrodynamics of the step and pool by a plain language.

As said, the work is of high quality, but I do have a major issue with the take home message of the article, which can be summarised with the last sentence of the discussion: "Based on the above arguments, we propose that the use of the mass transfer rate, k, should be dismissed in cases in which the heterogeneity of the flow field controls the fraction of mass evaded into the atmosphere, as in our step-and-pool configuration." I fully agree that gas transfer velocities are tricky to measure, highly variable in space and tricky to translate from one spatial scale to another.

Thank you for the comment and the general agreement on the main point of the paper.

There are also multiple methodologies for different use cases, see the review by Hall and Ulseth 2020 (WIRES) for a great overview.

The reference is already included in the reference list.

In this case, a reach-scale measure of the gas transfer velocity using a gas tracer, where a inert gas is injected upstream and the loss is quantified downstream with reaches of 50-150 meters for example, is a perfectly valid method to quantify gas exchange in a stream with steps and pools.  Spatial aggregation is indeed important, and how those reach scale measurements are translated to catchment scales remain an unanswered issue for instance. I will agree with the authors that for example, using chambers in a step and pool system is not a good method, but their claim is a bit overreaching.

The adoption of inert gas can allow us to quantify the whole gas exchange in a given portion of a watercourse, but not the internal patterns; however, the estimated k values do not necessarily depend on the mean physical quantities defining the reference reach (slope, velocity etc) because they are mostly related to the specific topography of the reach. Likely, we can upscale gas exchange in heterogeneous streams, as in those characterized by steps and pools, only by separating the gas evasion contributions deriving from characteristic geomorphic elements (steps, segments, cascades, etc), as shown in Botter et al. (2022). The estimation of the average value of k by the gas tracer in a 50-150 m of reach cannot allow one to recognize the role of these elements and, consequently, properly upscale and extrapolate the direct experimental measure. In the revised version of the paper we will be more cautious in proposing that k should be dismissed, and we will explain the main issues associated with methods that do not resolve the internal heterogeneity of reaches. We feel we have to thank the referee for his/her comment.

I will conclude this with an analogy to another physical and turbulent system. Temperature of a fluid is some kind of emergent property, which is related to the average movement of all molecules. You could use some great technology to track and quantify the movement of all particles, only to realise that the system is highly chaotic and heterogenous with a lot of eddies, indicating that is rally hard to quantify the movement of particles. Regardless, at a larger scale we have some other tools to estimate temperature of that system at larger scales that may be a simplification but work well enough. This study is a bit similar in this sense, it provides unique insight of the fine scale turbulent dynamics of water, suggests that is highly heterogeneous, but the link to the larger reach scale is a bit weak that may need some improvement or toning down in the text.

Thank you for your comment, which offers an interesting starting point for reflection. We think the analogy of the temperature can be useful to explain our point of view. Your preamble is correct but your reasoning does not directly face the question: "What is my temperature data for?". Whether the process of interest is linearly dependent on T, the use of this "average" of the molecules' movements works very well. On the contrary, non-linear processes involving the dynamics of the molecules are not captured by the averaged quantities. The gas exchanged between water and the atmosphere belongs to this second case, as the gas exchange and the mass transfer rate are non linear functions of the dissipated energy (see Botter et al., 2021). Of course, in practice, a perfect knowledge of the k patterns (or of the molecules' movements) is hardly feasible, and some average quantities are mandatory. However, this procedure determines small uncertainties only in nearly-homogenous conditions, i.e., when the k has the same order of magnitude along

the whole study reach. This is not the case for streams with step-and-pools where k may vary by some order of magnitude, and in which we need to isolate the contribution of k owing to the jet at a scale not detectable by in-field instruments..

However, the final statement of the paper can be toned down, and the criticisms of the adoption of k at the reach scale in high-energy streams will be better substantiated in the revised version of the paper.

Below I detail some minor issues I found in the text:

Abstract

-L10: maybe is cleaner to put k in parenthesis?

We will edit the abstract accordingly.

Introduction

In the first and second paragraph, the authors focus very fast on mountain streams. Step and pools are common in other landscapes outside mountains, so maybe it would help rising the generality of the article to discuss this more broadly outside mountain streams.

We will revise the mentioned paragraph according to your advice. We will include in the description other settings where steps and pools are common, as described in Chin and Whol (2005) and the cited studies therein.

Chin, A., & Wohl, E. (2005). Toward a theory for step pools in stream channels. *Progress in physical geography*, 29(3), 275-296.

Methods

-L121: The symbol of a bar with a dot above and one below may be unfamiliar to many readers of Biogeosciences. Define it in parentheses.

We will use a better symbol instead of ÷.

-L121-132:This whole paragraph is highly technical but still can be understood by a broader readership. This would be more likely with the support of more references as only Zappa et al 2007 is cited here. For example it would be helpful to have a citation after "Batchelor scale" (L124), "in its energy cascade (L126).

We will add these additional references as suggested (see list below). Further we will rephrase the paragraph to improve the readability for a general audience.

Esters, L., Landwehr, S., Sutherland, G., Bell, T. G., Saltzman, E. S., Christensen, K. H., ... & Ward, B. (2016). The relationship between ocean surface turbulence and air-sea gas transfer velocity: An in-situ evaluation. *In IOP Conference Series: Earth and Environmental Science* (Vol. 35, No. 1, p. 012005). IOP Publishing.

Batchelor, G. K. (1959). Small-scale variation of convected quantities like temperature in turbulent fluid Part 1. General discussion and the case of small conductivity. *Journal of fluid mechanics*, 5(1), 113-133.

Tennekes, H., & Lumley, J. L. (1972). *A first course in turbulence*, pp 257-264. MIT press.

Katul, G., & Liu, H. (2017). Multiple mechanisms generate a universal scaling with dissipation for the air-water gas transfer velocity. *Geophysical Research Letters*, 44(4), 1892-1898.

-L134: citation supporting this? Maybe Hall and Ulseth 2020 WIRES water?

Thank you for the suggestion. We will add the quotation in the revised text.

In addition to the one mentioned, we will include:

Hall Jr, R. O., & Madinger, H. L. (2018). Use of argon to measure gas exchange in turbulent<? xmltex\break?> mountain streams. *Biogeosciences*, *15*(10), 3085-3092.

Raymond, P. A., Zappa, C. J., Butman, D., Bott, T. L., Potter, J., Mulholland, P., ... & Newbold, D. (2012). Scaling the gas transfer velocity and hydraulic geometry in streams and small rivers. *Limnology and Oceanography: Fluids and Environments*, *2*(1), 41-53.

Ulseth, A. J., Hall Jr, R. O., Boix Canadell, M., Madinger, H. L., Niayifar, A., & Battin, T. J. (2019). Distinct air–water gas exchange regimes in low-and high-energy streams. *Nature Geoscience*, *12*(4), 259-263.

Maurice, L., Rawlins, B. G., Farr, G., Bell, R., & Gooddy, D. C. (2017). The influence of flow and bed slope on gas transfer in steep streams and their implications for evasion of $CO_2$. *Journal of Geophysical Research: Biogeosciences*, *122*(11), 2862-2875.

-L161: A brief explanation of this solubility coefficient?

The Ostwald solubility coefficient is the ratio between the volume of absorbed gas and the volume of absorbing liquid for a given condition of pressure and temperature. We will add this explanation in the revised version of the manuscript.

Results

-L177: "Provided by the code" might be better to say "provided by the model"

-L190 "Typo in "Negative values are observer" -> observed

Thanks. We will fix these minor issues.

-L205: It is unclear if it was one or two orders of magnitude. Might need to rephrase.

-L223: No need to say what you did or what you show in a figure. You can directly explain the observation and cite the figure in parenthesis.

We will rephrase the two sentences as suggested.

Discussion

-the discussion is very well written, despite the main comment that mostly concerns the discussion I have no small issues.

Thank you for the positive comment.

---

## Author Response (AR1)

**Response to the Associate Editor**

After having received reviews and reading your comments on those, I am happy to suggest publication of your article after minor revisions. I urge you to react carefully to the reviewer suggestions.

Many thanks for the positive evaluation of our work. We considered all the reviewers suggestions and implemented the required editing to the manuscript, as you can read in the following point-by-point rebuttal.

Two points seem particularly important to me and will require changes to the manuscript:

\*) Please improve the presentation of your work, in particular the various equations and mathematical derivations, so that your work may be accessible to a wider range of scientists from fields ranging from computational fluid dynamics to biogeochemistry and stream ecology.

According to your suggestion, we improved the presentation of the theoretical aspects by a plain language. In particular, we included the following brief forewords in Sections 2.3.1 and 2.3.2 to explain the physics of the models to calculate $k_T$ and $k_B$, respectively.

> Existing estimates of gas exchange in the presence of turbulent flows are based on the idea to link $k_T$ with some key characteristic hydrodynamics quantity (e.g., the energy dissipation). In mountain streams, with low water depths and high velocities, the theoretical schemes that are better suited to quantify $k_T$ are those based on small-eddies models (Moog and Jirka, 1999b). In this frame, the renewal of the free surface of the water column is led by the small-scale vortexes originated by the turbulence near the interface, which in turn are controlled by the rate of dissipation per mass unit of the turbulent kinetic energy, $\varepsilon$. According to this model, $k_T$ scales with $\varepsilon$, as follows

> In this work, while dealing with bubble-mediated gas exchange we focus on the so-called "kinematic bubble effect" (Liang et al., 2013). According to our approach, the amount of gas transferred from the water to the bubbles (or vice versa) is estimated using the following three-steps procedure: (A) first we determine the reference bubble lifetime, $T_B$ (i.e., the time spent by the bubble in the water column when it reaches the surface); (B) then, we calculate the gas transferred across the bubble interface according to the difference in concentration with the surrounding bulk flow during the bubble flow path; (C) finally, the total gas transfer velocity is estimated by considering the total amount of bubbles involved in the process.

Finally, we added the following additional references to better frame the hydrodynamic notions introduced

Batchelor, G. K. (1959). Small-scale variation of convected quantities like temperature in turbulent fluid Part 1. General discussion and the case of small conductivity. *Journal of fluid mechanics*, 5(1), 113-133.

Tennekes, H., & Lumley, J. L. (1972). *A first course in turbulence*, pp 257-264. MIT press.

Jähne, B., Heinz, G., & Dietrich, W. (1987). Measurement of the diffusion coefficients of sparingly soluble gases in water. *Journal of Geophysical Research: Oceans*, 92(C10), 10767-10776.

\*) Issues regarding spatial scale and your main conclusion were raised by both reviewers. Please reframe your conclusions in order to support meaningful translation of work across spatial scales in theory and future practical work. The main point of reviewers was that work at larger spatial scale is meaningful and needed and definitely not declared unvalid by your findings. Rather, your findings may help future work in supporting

appropriate and easier assessments of gas exchange. Bottomline: Please identify and strive for more constructive formulation of your findings.

Following the Reviewers observations, we reframed our conclusion toning down the limits of the use of spatially averaged k, indicating the potential solution to use k in the light of the findings of the present analysis. Specifically, the conclusion of the Discussion at L311-319, now reads

> Thus, the average value of $k$ across a step is not only unpredictable but also inevitably scale-dependent, i.e., the larger the water volume involved in the averaging, the lower the effective mass transfer velocity $k$ (Botter et al. 2021, 2022). As a consequence, aggregating $k$ across heterogeneous reaches might be inherently biased, and extrapolating observed reach-wise mass transfer rates could be appropriate only in a limited subset of reaches that share the same degree of heterogeneity of the reach where the experiments were performed. Based on the above arguments, we propose that the use of the mass transfer rate, $k$, should be used with caution when the heterogeneity of the flow field controls the fraction of mass evaded into the atmosphere, as in our step-and-pool configuration. In these cases, the mass transfer rate could be highly site-specific, and its extrapolation to different settings could be feasible only following a detailed analysis of the evasion produced by the constitutive geomorphic elements of the focus reach (including steps and cascades) (Botter et al., 2022).

I am looking forward to receiving your revised manuscript.

Regards, Gabriel Singer

**Response to Reviewer #1**

General comments

The manuscript entitled 'Local processes with global impact: unraveling the dynamics of gas evasion in a step-and-pool configuration' by Peruzzo et al. focuses on the mechanisms driving high gas exchange velocity in step-pool streams. The authors used an artificial step-pool flume to evaluate gas exchange by investigating the role hydrodynamics on energy dissipation. Gas exchange is driven by diffusivity but enhanced with turbulence and gas entrainment in bubbles. Turbulence and bubble-mediated gas exchange is not easy to separate, which the authors do here and is a key component of identifying the mechanism behind high gas-exchange in mountain streams. The authors found that gas exchange in the artificial step-pool was highly heterogenous. The greatest gas-exchange was at the spout – where the water dropped from the step to the pool. Also, under steady-stream conditions, the area of bubbles created from the spout varied spatially over time. Bubble mediated gas exchange varied and was 2.5 to 5-fold greater than turbulent gas exchange. Overall, the authors found that bubble mediated gas exchange dominated in the 'bubble' zone whereas turbulent mediated gas exchange dominated in the 'calmer' zones of the flume with both types of gas exchange being highly spatially heterogenous. This is likely driven by the strong spatial heterogeneity of energy dissipation.

The authors discuss the importance of taking spatial heterogeneity into account in these types of streams, which I agree given their findings.

Many thanks for the careful synopsis of our work and for the generally positive comments on our manuscript. All suggestions and queries of the referee have been carefully considered and most of them have been properly complied with, as detailed below.

However, I think scale is also highly dependent on how one might evaluate or estimate gas exchange. In field settings, we do rely on reach-scale gas exchange estimates in mountain streams – often from tracer gas experiments – as other methods are not feasible (i.e., night-time regression from continuous oxygen measurements, domes given the high turbulence do not often work – besides some of the work cited here, see also Hall Jr., Robert O., and Hilary L. Madinger. "Use of Argon to Measure Gas Exchange in Turbulent Mountain Streams." *Biogeosciences* 15, no. 10 (May 18, 2018): 3085–92. https://doi.org/10.5194/bg-15-3085-2018). The work here illustrated how heterogenous k can be in a 120cm2 flume. At some point, averaging does occur - even within this experiment. I would caution suggesting that reach scale metrics in mountain streams are not adequate (if I understood the discussion correctly) - but rather selecting the reach with the heterogeneity in mind would be a helpful step of estimating gas exchange in the field as we try to move towards some kind of scaling mechanism or even for any reach-specific studies.

We are aware that the in-field measures of k, which usually occur on the reach scale and constrain the possibility of catching the heterogeneity observed in the present analysis, are essential to determine gas exchange dynamics, and any averaging operation is, in any case, epistemologically inescapable. Nevertheless, the pattern of k observed is dependent on the local features of the channel and, thus, the use of the averaged k is appropriate only for the specific reach under investigation (or in a limited subclass of reaches presenting analogous conformations). We have better specified this point in the new version of the manuscript that at L311-315 now reads:

> Thus, the average value of $k$ across a step is not only unpredictable but also inevitably scale-dependent, i.e., the larger the water volume involved in the averaging, the lower the effective mass transfer velocity $k$ (Botter et al. 2021, 2022). As a consequence, aggregating $k$ across heterogeneous reaches might be inherently biased, and extrapolating observed reach-wise mass transfer rates could be appropriate only in a limited subset of reaches that share the same degree of heterogeneity of the reach where the experiments were performed.

Furthermore, we have added to the bibliography the suggested reference (L286).

Specific comments

L21: missing a closing paratheses after 'such as CO2 and N2O to the atmosphere)

Thank you for noting that. The text has been revised accordingly.

L36: missing 's' for 'see e.g. Cirpka et al. 1993'

The above typo mistake has been fixed.

L196: I appreciate quantifying the bubbles as bubble mediated gas exchange can be so much greater than turbulent driven k. The bubbles that were 'hardly feasible' to detect – would they not still have a significant effect on gas exchange significantly from turbulent driven k? On L264 it is discussed heterogenous size of the bubbles should not matter when estimating k. However, the diameter of the bubble is accounted for in equation 6. Perhaps as long as the bubbles are below 0.82 mm for radius, then bubble size should not matter as much?

Thank you for the valuable comment. The exchange of gas driven by bubbles is modeled as reported in Section 2.3.2. According to this model, for a given flux $q_B$, the contribution on $k_B$ of small-size bubbles can be potentially higher than that of the large-size bubbles. This may imply that the smallest (and not visible) bubble could be pivotal in the exchange process. However, the relevance (or not) of a specific size class of bubbles also depends on their mass fraction, which is proportional to the bubble size (and in any case would need to be determined experimentally). The quantification of the role of bubbles with different size in the gas exchange process is deferred to further work, and here it is assumed that the order of magnitude of the processes involved is properly captured if the size distribution of the bubbles is neglected and only the mean bubble diameter is considered. This point is further clarified in the new version of the manuscript as follows (L270-277).

> Some caution is needed when transposing this numerical result to real-world settings, as the outgassing driven by bubbles may depend not only on the step geometry, but also on the size and distribution of the bubbles (see Equations (6)-(8)). In our simulation, we have used a fixed bubble radius, which was set to represent the size of the visible bubbles observed during the experiment. Thus, we have neglected the role of the microbubbles, in the light of the fact that the mass fraction associated to small bubbles is expected to be negligible (Garrett et al., 2000). Although this analysis is thus based on a set of simplifying assumptions, we believe that the orders of magnitude of the underlying processes are properly captured by the model, as space-time heterogeneity in bubble size should not impact significantly the estimated flux of entrained air.

Garrett, C., Li, M., & Farmer, D. (2000). The connection between bubble size spectra and energy dissipation rates in the upper ocean. *Journal of physical oceanography*, 30(9), 2163-2171.

L202: 'The decrease in [energy dissipation] with the depth was more than linear'. I don't quite follow 'more than linear' - do the authors mean the relationship was not linear? I suggest expanding what kind of relationship was evident here for clarification.

According to the referee suggestion, we replaced "more than linear" with "superlinear".

L205: I don't quite follow 'the energy dissipation rate was one order (at least two orders)...' I suggest for clarity – if it was indeed 2-fold difference in energy dissipation between the two regions – state that it was two-fold.

The statement correctly referred to "one or two order of magnitude", which means 10- and 100-fold, respectively (see Figure 4).

L266: I absolutely agree that likely in mountain streams – when bubbles are present – bubble-mediated gas exchange drives k.  This work here is a great push forward to demonstrate bubble-vs-turbulent driven gas exchange by quantifying kb vs kt.  However, this is not the first work to propose kb may dominate gas exchange in highly turbulent streams, which I suggest be cited appropriately.

Of course, we agree that other works suggest the prominent role of the bubble to determine k in high-energy streams; we have expanded the references to acknowledge previous works that suggested the key role of bubbles in gas exchange at the water air interface as indicated (L281-282).

**Response to Reviewer #2**

The article by Peruzzo et al explores the gas exchange dynamics in a step-and-pool situation, which is a common geomorphic feature in many streams with some relieve like mountain streams. To do this they use a lab setup, with a unique monitoring set that measures water velocity and turbulence in multiple directions and at a high spatial and temporal resolution. This is something I have rarely seen so the authors can explore experimentally at a great resolution the drivers of gas exchange in waters. They can also separate bubble mediated gas exchange which is important and often ignored.

Thank you for the nice summary of our work.

They find a high spatial in the gas exchange, both turbulence and bubble mediated, with a very small area accounting for most of the gas exchange with the atmosphere. The article is well written and figures of high quality…

Thank you for the positive assessment of our manuscript.

…but the technical level of some of the section may be hard to follow for not only the average reader of "Biogeosciences", but also for someone who works with gas exchange in rivers without a hydrologic engineering background like me. With this I provide some suggestion to clarify some terms, notations, and provide citations on those technical aspects of the work.

Thank you for your comment. We worked to improve the readability for the average readership of Biogeosciences starting from your suggestions. In particular, we revised Section 2.3 including additional references and explaining the most technical aspects related to the hydrodynamics of the step and pool by a plain language as you suggested.

Specifically at L119-124 the revised version of the manuscript now reads

> Existing estimates of gas exchange in the presence of turbulent flows are based on the idea to link $k_T$ with some key characteristic hydrodynamics quantity (e.g., the energy dissipation). In mountain streams, with low water depths and high velocities, the theoretical schemes that are better suited to quantify $k_T$ are those based on small-eddies models (Moog and Jirka, 1999b). In this frame, the renewal of the free surface of the water column is led by the small-scale vortexes originated by the turbulence near the interface, which in turn are controlled by the rate of dissipation per mass unit of the turbulent kinetic energy, $\varepsilon$. According to this model, $k_T$ scales with $\varepsilon$, as follows

And at L153-158

> In this work, while dealing with bubble-mediated gas exchange we focus on the so-called "kinematic bubble effect" (Liang et al., 2013). According to our approach, the amount of gas transferred from the water to the bubbles (or vice versa) is estimated using the following three-steps procedure: (A) first we determine the reference bubble lifetime, $T_B$ (i.e., the time spent by the bubble in the water column when it reaches the surface); (B) then, we calculate the gas transferred across the bubble interface according to the difference in concentration with the surrounding bulk flow during the bubble flow path; (C) finally, the total gas transfer velocity is estimated by considering the total amount of bubbles involved in the process.

As said, the work is of high quality, but I do have a major issue with the take home message of the article, which can be summarised with the last sentence of the discussion: "Based on the above arguments, we

propose that the use of the mass transfer rate, k, should be dismissed in cases in which the heterogeneity of the flow field controls the fraction of mass evaded into the atmosphere, as in our step-and-pool configuration." I fully agree that gas transfer velocities are tricky to measure, highly variable in space and tricky to translate from one spatial scale to another.

Thank you for the comment and the general agreement on the main point of the paper.

There are also multiple methodologies for different use cases, see the review by Hall and Ulseth 2020 (WIRES) for a great overview.

The reference is already included in the reference list.

In this case, a reach-scale measure of the gas transfer velocity using a gas tracer, where a inert gas is injected upstream and the loss is quantified downstream with reaches of 50-150 meters for example, is a perfectly valid method to quantify gas exchange in a stream with steps and pools. Spatial aggregation is indeed important, and how those reach scale measurements are translated to catchment scales remain an unanswered issue for instance. I will agree with the authors that for example, using chambers in a step and pool system is not a good method, but their claim is a bit overreaching.

The adoption of inert gas can allow us to quantify the whole gas exchange in a given portion of a watercourse, but not the internal patterns; however, the estimated k values do not necessarily depend on the mean physical quantities defining the reference reach (slope, velocity etc) because they are mostly related to the specific topography of the reach. Likely, we can upscale gas exchange in heterogeneous streams, as in those characterized by steps and pools, only by separating the gas evasion contributions deriving from characteristic geomorphic elements (steps, segments, cascades, etc), as shown in Botter et al. (2022). The estimation of the average value of k by the gas tracer in a 50-150 m of reach cannot allow one to recognize the role of these elements and, consequently, properly upscale and extrapolate the direct experimental measure. In the revised version of the paper we are more cautious in proposing that k should be dismissed, and we explained the main issues associated with methods that do not resolve the internal heterogeneity of reaches (L311-319). We feel we have to thank the referee for his/her comment.

I will conclude this with an analogy to another physical and turbulent system. Temperature of a fluid is some kind of emergent property, which is related to the average movement of all molecules. You could use some great technology to track and quantify the movement of all particles, only to realise that the system is highly chaotic and heterogenous with a lot of eddies, indicating that is really hard to quantify the movement of particles. Regardless, at a larger scale we have some other tools to estimate temperature of that system at larger scales that may be a simplification but work well enough. This study is a bit similar in this sense, it provides unique insight of the fine scale turbulent dynamics of water, suggests that is highly heterogeneous, but the link to the larger reach scale is a bit weak that may need some improvement or toning down in the text.

Thank you for your comment, which offers an interesting starting point for reflection. We think the analogy of the temperature can be useful to explain our point of view. Your preamble is correct but your reasoning does not directly face the question: "What is my temperature data for?". Whether the process of interest is linearly dependent on T, the use of this "average" of the molecules' movements works very well. On the contrary, non-linear processes involving the dynamics of the molecules are not captured by the averaged quantities. The gas exchanged between water and the atmosphere belongs to this second case, as the gas exchange and the mass transfer rate are non linear functions of the dissipated energy. Of course, in practice, a perfect knowledge of the k patterns (or of the molecules' movements) is hardly feasible, and some average quantities are mandatory. However, this procedure determines small uncertainties only in nearly-homogenous conditions, i.e., when the k has the same order of magnitude along the whole study reach. This

is not the case for streams with step-and-pools where k may vary by some order of magnitude, and in which we need to isolate the contribution of k owing to the jet at a scale not detectable by in-field instruments.

In response to this comment we have decided to tone down the last part of the discussion / conclusion, and the limits of the use of spatially averaged reachwise k in high-energy streams were better substantiated in the revised version of the paper (L315-319).

> Based on the above arguments, we propose that the use of the mass transfer rate, *k*, should be used with caution when the heterogeneity of the flow field controls the fraction of mass evaded into the atmosphere, as in our step-and-pool configuration. In these cases, the mass transfer rate could be highly site-specific, and its extrapolation to different settings could be feasible only following a detailed analysis of the evasion produced by the constitutive geomorphic elements of the focus reach (including steps and cascades) (Botter et al., 2022).

Below I detail some minor issues I found in the text:

Abstract

-L10: maybe is cleaner to put k in parenthesis?

We edited the abstract accordingly.

Introduction

In the first and second paragraph, the authors focus very fast on mountain streams. Step and pools are common in other landscapes outside mountains, so maybe it would help rising the generality of the article to discuss this more broadly outside mountain streams.

We revised the mentioned paragraph according to your advice. The new version of the manuscript now reads (L33-34):

> Recent studies evidenced that local steps are important hotspots of gas evasion, to the point that in all the settings where steps are distinctive of the stream's morphology (e.g., in high-energy rivers or in streams originated by glacial processes), they control the overall network-scale outgassing (Vautier et al., 2020; Botter et al., 2022).

Methods

-L121: The symbol of a bar with a dot above and one below may be unfamiliar to many readers of Biogeosciences. Define it in parentheses.

We used  – instead of ÷.

-L121-132:This whole paragraph is highly technical but still can be understood by a broader readership. This would be more likely with the support of more references as only Zappa et al 2007 is cited here. For example it would be helpful to have a citation after "Batchelor scale" (L124), "in its energy cascade (L126).

We added these additional references as suggested (see list below).

Batchelor, G. K. (1959). Small-scale variation of convected quantities like temperature in turbulent fluid Part 1. General discussion and the case of small conductivity. *Journal of fluid mechanics*, 5(1), 113-133.

Tennekes, H., & Lumley, J. L. (1972). *A first course in turbulence*, pp 257-264. MIT press.

Jähne, B., Heinz, G., & Dietrich, W. (1987). Measurement of the diffusion coefficients of sparingly soluble gases in water. *Journal of Geophysical Research: Oceans*, 92(C10), 10767-10776.

-L134: citation supporting this? Maybe Hall and Ulseth 2020 WIRES water?

Thank you for the suggestion. We will add the quotation in the revised text.

In addition to the one mentioned by the referee, we will include:

Hall Jr, R. O., & Madinger, H. L. (2018). Use of argon to measure gas exchange in turbulent<? xmltex\break?> mountain streams. *Biogeosciences*, *15*(10), 3085-3092.

Raymond, P. A., Zappa, C. J., Butman, D., Bott, T. L., Potter, J., Mulholland, P., ... & Newbold, D. (2012). Scaling the gas transfer velocity and hydraulic geometry in streams and small rivers. *Limnology and Oceanography: Fluids and Environments*, *2*(1), 41-53.

Ulseth, A. J., Hall Jr, R. O., Boix Canadell, M., Madinger, H. L., Niayifar, A., & Battin, T. J. (2019). Distinct air–water gas exchange regimes in low-and high-energy streams. *Nature Geoscience*, *12*(4), 259-263.

-L161: A brief explanation of this solubility coefficient?

The Ostwald solubility coefficient is the ratio between the volume of absorbed gas and the volume of absorbing liquid for a given condition of pressure and temperature. We will add this explanation in the revised version of the manuscript.

We added this definition at L171-172 of the revised manuscript, which now reads

> where $\beta$ is the Ostwald solubility coefficient, which is the ratio between the volume of absorbed gas and the volume of absorbing liquid for a given condition of pressure and temperature.

Results

-L177: "Provided by the code" might be better to say "provided by the model"

We replaced "code" with "model" as suggested.

-L190 "Typo in "Negative values are observer" -> observed

Thanks. We will fix these minor issues.

-L205: It is unclear if it was one or two orders of magnitude. Might need to rephrase.

We rephrased the sentence as follows (L215-217).

> the energy dissipation rate was one order and at least two orders of magnitude larger than that estimated in the backwater region and downstream of the cascade, respectively.

-L223: No need to say what you did or what you show in a figure. You can directly explain the observation and cite the figure in parenthesis.

We erased the sentence and cite the Figure 4b in the next paragraph.

Discussion

-the discussion is very well written, despite the main comment that mostly concerns the discussion I have no small issues.

Thank you for the positive comment.